# Thermally active nanoparticle clusters enslaved by engineered domain wall traps

Pietro Tierno ⬤ 1,2,3✉, Tom H. Johansen4,5 & Arthur V. Straube ⬤ 6✉

The stable assembly of fluctuating nanoparticle clusters on a surface represents a technological challenge of widespread interest for both fundamental and applied research. Here we demonstrate a technique to stably confine in two dimensions clusters of interacting nanoparticles via size-tunable, virtual magnetic traps. We use cylindrical Bloch walls arranged to form a triangular lattice of ferromagnetic domains within an epitaxially grown ferrite garnet film. At each domain, the magnetic stray field generates an effective harmonic potential with a field tunable stiffness. The experiments are combined with theory to show that the magnetic confinement is effectively harmonic and pairwise interactions are of dipolar nature, leading to central, strictly repulsive forces. For clusters of magnetic nanoparticles, the stationary collective states arise from the competition between repulsion, confinement and the tendency to fill the central potential well. Using a numerical simulation model as a quantitative map between the experiments and theory we explore the field-induced crystallization process for larger clusters and unveil the existence of three different dynamical regimes. The present method provides a model platform for investigations of the collective phenomena emerging when strongly confined nanoparticle clusters are forced to move in an idealized, harmonic-like potential.

[1] Departament de Física de la Matèria Condensada, Universitat de Barcelona, 08028 Barcelona, Spain. [2] Institut de Nanociència i Nanotecnologia, Universitat de Barcelona, Barcelona, Spain. [3] Universitat de Barcelona Institute of Complex Systems (UBICS), Universitat de Barcelona, Barcelona, Spain. [4] Department of Physics, University of Oslo, P.O. Box 1048 Blindern, 0316 Oslo, Norway. [5] Institute for Superconducting and Electronic Materials, University of Wollongong, Northfields Avenue, Wollongong, NSW 2522, Australia. [6] Zuse Institute Berlin, Takustraße 7, 14195 Berlin, Germany. ✉email: ptierno@ub.edu; straube@zib.de

S table localization of nanoparticle clusters on a surface may lead to several technological developments in different research fields, ranging from drug delivery[1,2], to microfluidics[3], optics[4], and photonics[5]. Understanding the adsorption and mobility of nanoparticles on a substrate is also crucial in different surface-based technologies such as friction[6,7] and heterogeneous catalysis[8]. For example, localized impurities as inorganic fullerene-like nanoparticles showed excellent lubricating properties on layered materials, with the enhancement of sliding friction and exfoliation-material transfer[9]. In addition, investigating the dynamics of confined nanoparticle clusters is also important from a theoretical point of view. These clusters represent a simple yet nontrivial model system for studying multi-body effects in condensed matter physics, and understanding their mesoscopic dynamics may shed light on other systems on different length- and timescales. Examples span from the arrangements of electrons in a parabolic potential, where they form a Wigner crystal[10], to the dynamics of ions in a lateral optical confinement[11].

In colloidal science, the formation of clusters of microscale particles via isotropic, attractive interactions and their complex transition pathways have been investigated in two[12] and in three dimensions[13,14]. More recently, the effect of active perturbations on the assembly pathways has been reported with levitated granular particles assembled by acoustic forces.[15] These works demonstrated that current experimental techniques allow manipulation and control of cluster at the colloidal length scale and above. However, applying similar approaches to investigate clusters of magnetic nanoparticles remains a challenging task. At smaller length scales, large field gradients are required to overcome thermal fluctuations and provide a stable trapping site for nanoparticles. Some successful approaches have been recently proposed based on the use of plasmonic landscapes[16,17] or hard-wall confinements[18,19]. Most of such techniques require the use of lithographic patterns composed of fixed microstructures which may influence the dynamics due to steric interactions and cannot be easily changed by external control.

Here we demonstrate the stable trapping and control of thermally active clusters of magnetic nanoparticles in solution by using extended circular traps made of magnetic Bloch walls. These domain walls generate strong and tunable magnetic gradients which induce the assembly of nanoparticles into fluctuating clusters in two dimensions (2D). The research on controlled motion of domain walls in magnetic thin films is currently pushing the limit of magnetic data storage technology and is also providing applications in logic devices[20,21], spintronics[22], nanowires[23,24], and ultracold atoms[25]. We use these nanoscale entities to trap and control soft magnetic nanoparticles, as an alternative approach to optical tweezers[26] or dielectrophoretic traps[27].

## Results

**Experimental system**. We realize cylindrical ferromagnetic domains, or magnetic "bubbles", by using a single crystal, uniaxial ferrite garnet film (FGF), see Fig. 1a–c. The FGF is grown via dipping liquid phase epitaxy on a ⟨111⟩ oriented single crystal gadolinium gallium garnet ($Gd_3Ga_5O_{12}$)[28], and has the composition $Y_{2.5}Bi_{0.5}Fe_{5-q}Ga_qO_{12}$ ($q = 0.5-1$). The grown film displays a labyrinth pattern of stripe domains with alternating perpendicular magnetization vector. The domains are separated by ~20 nm thick Bloch walls (BWs) where the magnetization vector rotates by 180° in the $(x, z)$ plane. To transform the stripe pattern into a triangular lattice of magnetic bubbles, we anneal the lattice by keeping it for ~15 min to a high frequency (0.5 kHz)

magnetic field of amplitude $B_z = 3$ mT. After switching off the field, the FGF displays a triangular lattice of cylindrical ferromagnetic domains with uniform magnetization, lattice constant $a = 11.8\,\mu m$ and diameter $D = 8.8\,\mu m$. The size of the magnetic bubbles can be tuned by an external field perpendicular to the film, $\mathbf{B}_{ext} = B_z\hat{\mathbf{z}}$. When the field is parallel (antiparallel) to the bubble magnetization, it increases (decreases) the radius of the cylindrical domains. From the analysis of the experimental data, Fig. 1d, we extract a saturation magnetization of $B_s = 21.3$ mT (critical field $B_c = 11.4$ mT[29]). More technical details can be found in "Methods". We also note that recently a strong edge stray field has been generated by an array of regular nanorod assembled in a triangular lattice[30]. However, the presence of the oppositely magnetized surrounding film impedes the formation of localized vortices in our system.

**Nanoparticle trapping**. Above the magnetic lattice we deposit a water dispersion of paramagnetic nanoparticles with diameter $d = 360$ nm (Microparticles GmbH), and doped with ~40% by weight of iron oxide. Before placing the particles, we use soft lithography to cover the FGF with a 1 $\mu m$ thin layer of a polymer film to avoid sticking due to magnetic attraction, see "Methods" for more details. Due to density mismatch, the nanoparticles sediment in water reaching the FGF surface. As shown in Fig. 1c, see also Video S1 in the Supporting Information, the particles are effectively 2D confined within the cylindrical domains, and can only exchange position by moving within the $(x, y)$ plane, but not along the perpendicular direction.

The BW trapping at the center of the magnetic domains could be explained by calculating the magnetostatic potential generated by the stray field of the film, $\mathbf{B}_{stray}$. The energy of one paramagnetic particle at an elevation $z$ above such lattice is given by $U_1(\mathbf{r}) = -\upsilon\chi\mathbf{B}^2(\mathbf{r})/(2\mu_0)$, see "Methods". Here, $\mathbf{B} = \mathbf{B}_{ext} + \mathbf{B}_{stray}$ is the total magnetic field, $\upsilon$ the particle volume, $\chi \sim 2$ the effective magnetic volume susceptibility[31], and $\mu_0 = 4\pi \times 10^{-7}$ H/m is the magnetic permeability of the medium (water). To determine the energy landscape $U_1(x, y)$, we calculate numerically $\mathbf{B}_{stray}$ by summing up a two-dimensional triangular lattice of cylindrical ferromagnetic domains, see "Methods" for more details. Figure 1e shows the result for zero external field ($B_z = 0$) at the particle elevation of $z = 1.3\,\mu m$. The energy landscape displays a triangular lattice of radially symmetric potential wells that are centered at the locations of the magnetic bubbles. Further, from the calculations we find that when the applied field is parallel to the bubble magnetization ($B_z > 0$), these potential wells become deeper, while these minima disappear when the field is antiparallel to the bubble magnetization ($B_z < 0$); we will come back to this effect later.

**Single-particle fluctuations**. We proceed to determine the effective shape of the confining potential from the single-particle fluctuations. As shown in Fig. 2a, a trapped nanoparticle performs a confined, angle-independent diffusive motion relative to the center of the magnetic bubble at the origin. The external field $B_z$ increases the bubble area (red circle), and makes the potential stiffer, which induces a stronger confinement (red trajectory). This effect is confirmed in Fig. 2b, where from the particle trajectory we extract the effective confining potential $U_1(r)$, applying the standard Boltzmann distribution[32]. Even if the real trapping potential displays a complex shape (see Fig. 1e), we find that the nanoparticle explores only the central well staying away from the boundaries where the Bloch walls are located. In such region, the potential well can be well approximated with a harmonic function,

$$U_1(r) = \frac{1}{2}k_e r^2, \tag{1}$$

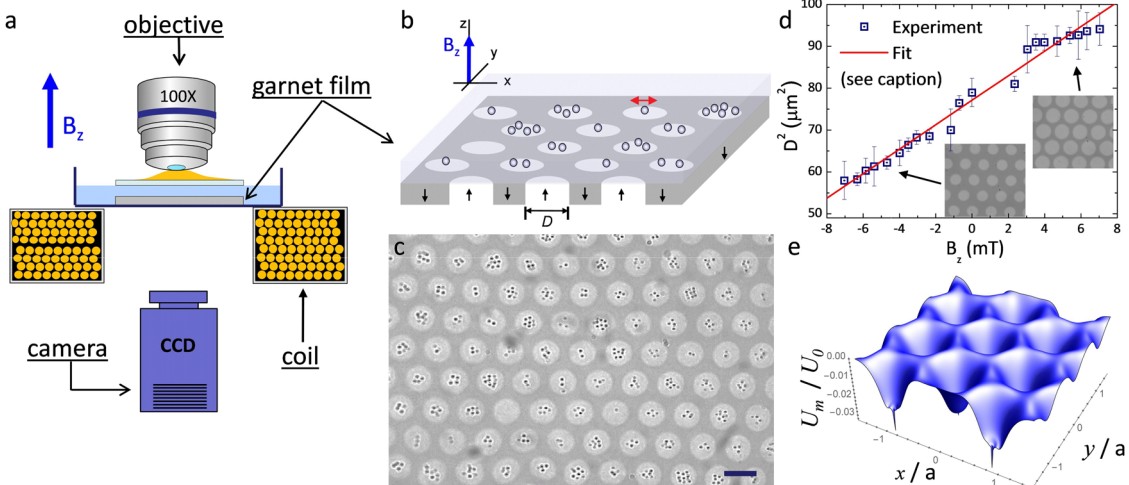

**Fig. 1 Magnetic bubble lattice. a** Schematic of the experimental setup used to visualize and control the Ferrite Garnet Film (FGF). **b** Detailed sketch of the FGF with magnetic bubble domains filled by different numbers of paramagnetic nanoparticles. The external magnetic field $\mathbf{B}_{ext} = B_z\hat{\mathbf{z}}$ is applied perpendicular to the film (z axis). **c** Polarization microscope image of trapped nanoparticles (of diameter $d = 360$ nm). The magnetic bubble domains are visible due to the polar Faraday effect. Scale bar is 10 $\mu$m, see also VideoS1 in the Supporting Information. **d** Square of the bubble diameter $D^2$ versus applied field $B_z$. Scattered points are experimental data while continuous line is a linear fit according to $D^2 = 4a^2[(B_z/B_s + 1)\sin(\pi/3)/(2\pi)]$ (see "Methods"), from which we extract the lattice constant $a = 11.81 \pm 0.02$ $\mu$m and the saturation magnetization $B_s = 21.3 \pm 0.3$ mT. Error bars in $D^2$ are obtained from the statistical average of different measurments. Insets show images of the magnetic domains. **e** Three-dimensional view of the magnetostatic potential $U_1$ calculated at an elevation $z = 1.3$ $\mu$m and for $B_z = 0$ mT. The $(x, z)$ positions are rescaled by $a$, while the potential $U_1$ is rescaled by the parameter $U_0 = \chi\pi d^3 B_s^2/(12\mu_0)$, see text for the values of $\mu_0$, $\chi$, and $d$.

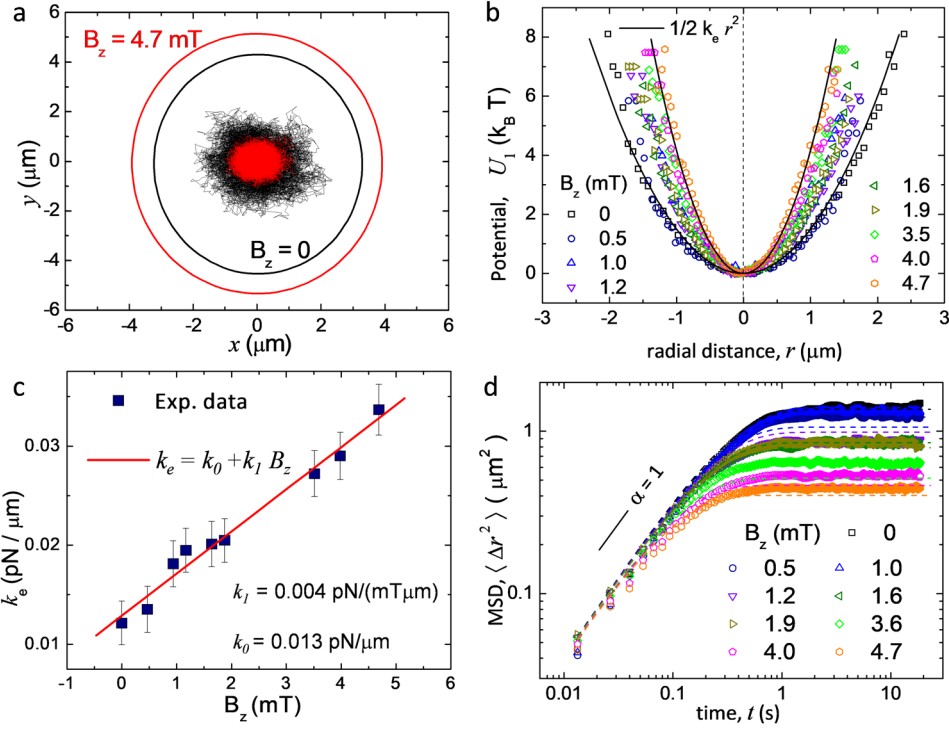

**Fig. 2 Single-particle dynamics. a** Two trajectories of one paramagnetic nanoparticle trapped above a magnetic bubble for $B_z = 0$ mT (black line) and $B_z = 4.7$ mT (red line), the circles indicate the corresponding locations of the Bloch wall. **b** Magnetic potential $U_1(r)$ calculated from the particle fluctuations for different values of the applied field $B_z$. Scattered points are experimental data, continuous lines are fit using a simple harmonic approximation, $U_1(r) = k_e r^2/2$. **c** Spring constant $k_e$ of the potential measured for different magnetic field $B_z$. Continuous red line denotes linear fit with $k_e = k_0 + k_1 B_z$, with $k_0 = 0.0129 \pm 0.0012$ pN/$\mu$m and $k_1 = 0.0043 \pm 0.0005$ pN/(mT$\mu$m). The error bars in the data result from the statistical average of different measurments. **d** Log–log plot of the mean-square displacement (MSD) for different applied magnetic fields. Continuous lines are fits obtained by using the theoretical model, see Eq. (2) and text.

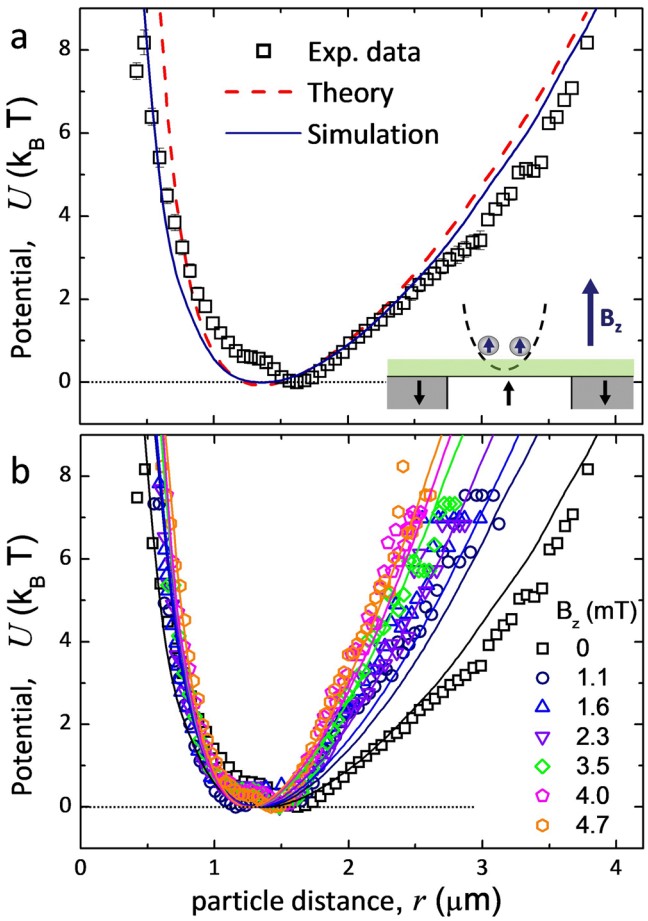

**Fig. 3 Total interaction potential. a** Total interaction potential $U(r)$ between two trapped nanoparticles evaluated from the distribution of the interparticle positions (symbols for experiment and continuous blue line for simulations) and from the formula $U(r) \approx 2U_1(r/2) + U_2(r)$ (theory, red dashed line) versus the interparticle distance $r$ for $B_z = 0$ mT. Small inset shows an image of a pair of confined nanoparticles interacting repulsively due to the parallel induced magnetic moments. **b** The potential $U(r)$ versus $r$ for different amplitudes $B_z$. Symbols and continuous lines refer to the experiment and simulations, respectively. The horizontal dotted line is guide to the eye for $U = 0$.

where $k_e$ is the effective spring constant. We note that exactly such radial dependence comes out naturally from our theoretical model when considering the close proximity of a magnetic bubble, see "Methods". Further, Fig. 2b shows that increasing the applied field makes the potential effectively stiffer and thus $k_e$ grows with $B_z$. Here, our model predicts a linearly growing function $k_e(B_z)$ (see "Methods"), which is in perfect agreement with the experiment. The corresponding experimentally measured values of the spring constant, extracted from different potential distributions, are shown in Fig. 2c. As expected, the potential stiffness varies linearly with the field amplitude, and we measure a maximum value of $k_e = 0.034$ pN/μm for $B_z = 4.7$ mT, which corresponds to a bubble of diameter $D = 9.5$ μm.

The particle dynamics can be further characterized by the mean-square displacement (MSD) given by, $\langle \Delta r^2 \rangle(t) = \langle (\mathbf{r}(t + t') - \mathbf{r}(t'))^2 \rangle$, where $\mathbf{r}(t) = (x(t), y(t))$ is the particle coordinate measured relative to the center of the bubble, $t$ is the lag time, and $\langle \ldots \rangle$ a statistical average, which we performed over ~15 independent experiments. The long-time behavior of the MSD can be described by a power law, $\langle \Delta r^2 \rangle(t) \sim t^\alpha$ with an exponent $\alpha$ which is used to distinguish

between normal ($\alpha = 1$) and anomalous ($\alpha \neq 1$) diffusion. In a harmonic trap, the particle dynamics is described by an overdamped Langevin equation, $\zeta \dot{\mathbf{r}} = -k_e \mathbf{r} + \boldsymbol{\xi}(t)$, which follows directly from our model, see "Methods". Here, $\zeta$ is the particle friction coefficient, and $\xi$ is a random force that describes thermal fluctuations. This equation is used to arrive at the 2D time-dependent MSD[33],

$$\langle \Delta r^2 \rangle(t) = \frac{2k_B T}{k_e} \left[ 1 - \exp\left( -\frac{2k_e t}{\zeta} \right) \right] , \quad (2)$$

which describes a crossover at $t \simeq \tau = \zeta/2k_e$ from free diffusion at small times ($t \ll \tau$), $\langle \Delta r^2 \rangle(t) \simeq 4D_0 t$ with $D_0 = k_B T/\zeta = 1.04$ μm²/s to confined Brownian motion, $\langle \Delta r^2 \rangle(t) = 2k_B T/k_e$ at long times ($t \gg \tau$). The long-time limit of this equation provides an independent way to measure the effective spring constants.

When using Eq. (2) to plot all the MSD curves, Fig. 2d, we detect slight differences in the values of spring constants $k_e$ compared to those determined from the potentials in Fig. 2c. This is not unexpected because the nanoparticle subjected to thermal noise explores the weakly anharmonic nature of the genuine confining potential and small deviations relative to the harmonic model are captured unequally well by the different approaches. As shown in "Methods", the account of weak anharmonicity of our magnetic trap keeps valid the harmonic model, Eq. (1), with an effectively suppressed spring constant, $k_e - \Delta k_e$, where the small correction $\Delta k_e \propto k_B T > 0$. Although accounting for such slight shift in $k_e$ allows us to achieve a better quantitative agreement for MSDs shown in Fig. 2d, it remains otherwise unessential for our study and therefore we neglect $\Delta k_e$ further.

**Pair interaction and collective states.** The circular magnetic traps can be used to enclose pairs of interacting nanoparticles and measure the corresponding pairwise interaction potential $U_2$ from the distribution statistics. As predicted by our theoretical model (see "Methods"), such interactions arise from the magnetic moments induced in paramagnetic nanoparticles by both the stray field of the FGF $\mathbf{B}_{stray} \approx 3e^{-\kappa z} B_s \hat{\mathbf{z}}$ and external field $\mathbf{B}_{ext} = B_z \hat{\mathbf{z}}$, where $\kappa \propto a^{-1}$ is a damping exponent. Note that the total field $\mathbf{B} = \mathbf{B}_{stray} + \mathbf{B}_{ext}$ and hence the induced dipoles remain always perpendicular to the confining plane, for any external field $B_z$, as shown in the schematic in Fig. 3a, including the case $B_z = 0$ mT. This naturally suggests that the pairwise dipolar interactions are central and strictly repulsive,

$$U_2(r) = \frac{\gamma}{r^3} , \quad (3)$$

with a field-dependent strength $\gamma = \gamma(B_z)$.

We show the validity of this hypothesis first for the case $B_z = 0$ mT, see Fig. 3a. In the presence of the external potential, the distribution of particle positions is dictated by the balance between the external and the repulsive dipolar forces. For a pair of particles with the displacements $r_1$ and $r_2$ from the trap center and interparticle distance $r$, we measure their total potential $U = U_1(r_1) + U_1(r_2) + U_2(r)$. A good analytic approximation for the total potential is given by $U(r) \approx 2U_1(r/2) + U_2(r)$, which is compared with $U(r)$ evaluated directly from the distribution of interparticle distances; more details are given in "Methods". We note that the experimental data, approximate formula and results of simulations are in good quantitative agreement, see Fig. 3a. However, as expected $U(r)$ computed from the particle distribution exhibits a slightly better correspondence to the experimental data than the prediction from the analytic formula with the same fitting parameters. Therefore, we stick to the more accurate representation when setting the mapping between the model and the experiments. Application of the external field, $B_z$, induces a

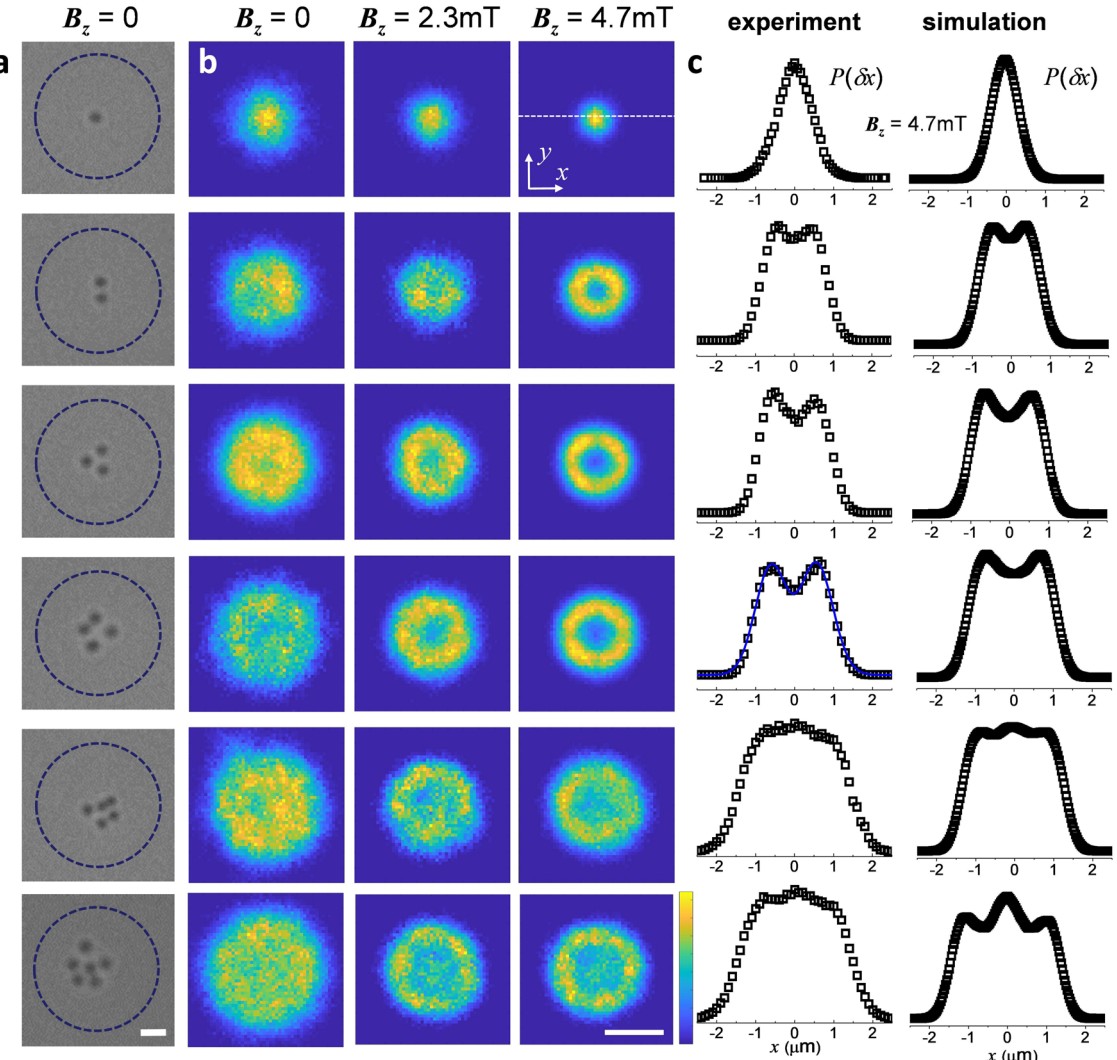

**Fig. 4 Stationary collective states in the magnetic trap. a** Sequence of optical microscope images showing different confined nanoparticles (from top to bottom) for $B_z = 0$ mT. The superimposed dashed line indicates the location of the Bloch wall, scale bar is $2\,\mu$m. **b** Two-dimensional density distribution of the particle positions projected onto the $(x, y)$ plane. High (low) probabilities correspond to yellow (blue) regions. Scale bar at the bottom is $2\,\mu$m. **c** Probability distributions $P(\delta x)$ of the particle displacement along the $x$-axis from experiments (left column) and numerical simulation (right column).

visible narrowing of the potential well: notice not only the right part of the graphs in Fig. 3b, but also steeper repulsion tails with the growth in $B_z$, as can be observed from the left part of the graphs. Fitting the simulation data to the experiment for $U(r)$ fixes the field-dependent strength $\gamma(B_z)$ of the pairwise repulsive interactions, $U_2(r)$.

When increasing the number of nanoparticles in the magnetic trap, the repulsive colloids are forced to coexist and they compete for filling the central minimum. Since the confining potential and interparticle interaction potential are now completely specified by Eqs. (1) and (3) with the parameters extracted from the experiment, we construct a Brownian dynamics simulation model. Combining these deterministic forces with irregular forces caused by thermal fluctuations, we arrive at the Langevin equations for $N$ interacting particles (see "Methods"),

$$\zeta \frac{d\mathbf{r}_i(t)}{dt} = -k_e \mathbf{r}_i + 3\gamma \sum_{j>i} \frac{\hat{\mathbf{r}}_{ij}}{r_{ij}^3} + \boldsymbol{\xi}_i(t) \,, \qquad (4)$$

used to rationalize the experiments and to characterize the emerging collective states. The evolution of particle $i$ with the position $\mathbf{r}_i = (x_i(t), y_i(t))$ is determined by the restoring, repulsion and thermal

noise forces, as given by the three terms in the right-hand side of Eq. (4), respectively.

We will now report collective states for an increasing number $N = 1, \dots, 6$ of particles confined to the trap, as shown in Fig. 4a for $B_z = 0$ mT. In Fig. 4b we show the stationary spatial density distribution $\rho(x, y)$ at different fields $B_z$ ranging from 0 to 4.7 mT. The corresponding one-dimensional displacement distributions $P(\delta x)$ obtained by cutting $\rho(x, y)$ along one direction ($y = 0$) are presented in Fig. 4c, as drawn from experiments (left column) and simulations (right). Both experiments and simulations show a very good agreement in describing the particle behavior within the trap. Under zero field (first column Fig. 4b) the density field remains highly localized only for a single particle, $N = 1$. Already for two particles, $N = 2$, the density distribution significantly flattens across the trapping region and each position within the magnetic trap has almost the same probability to be visited. With the further growth in $N$, this feature becomes more pronounced: the density distribution broadens, while staying nearly uniform in the most part of the trap. Note that although the particles repel, tending to keeping apart and filling larger regions, the repulsion strength remains relatively weak to break the quasi-uniform structure of $\rho(x, y)$.

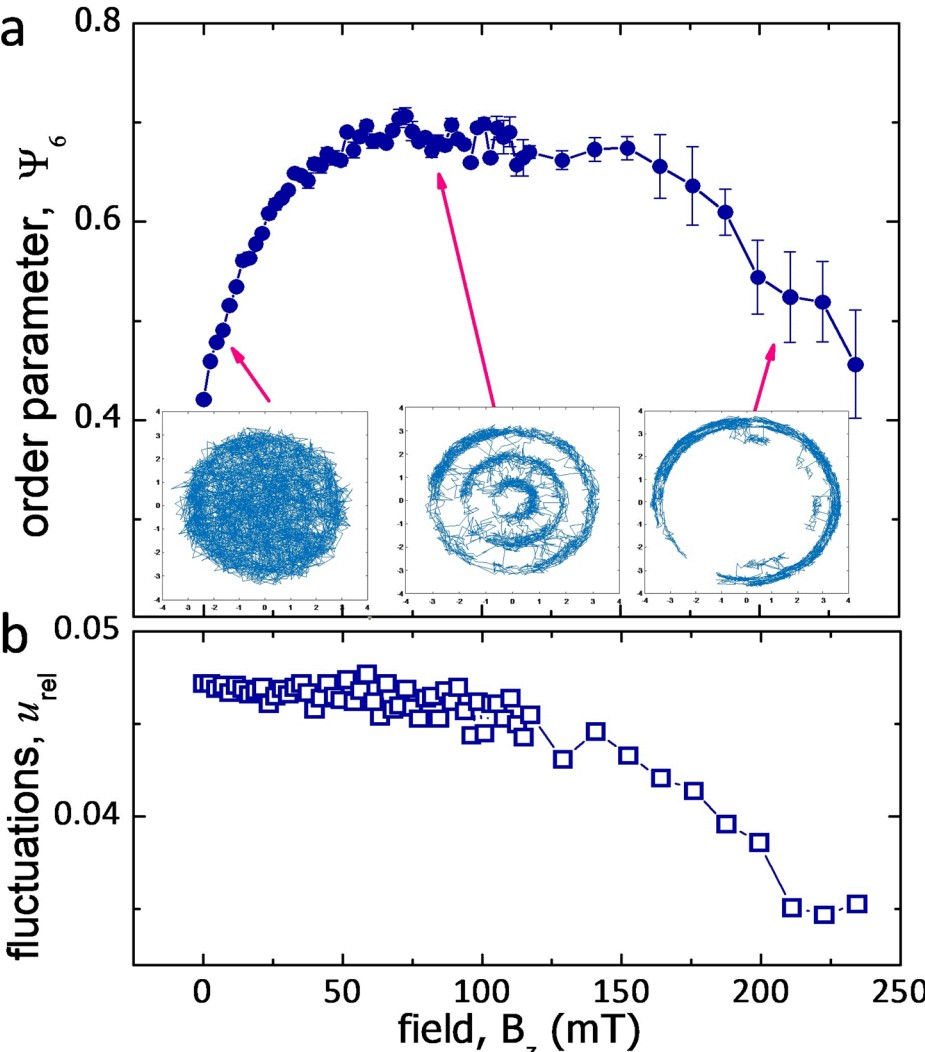

**Fig. 5 Field-induced crystallization process. a** Average orientational bond order parameter $\langle \Psi_6 \rangle$ versus amplitude $B_z$ of the applied field for a $N = 29$ nanoparticle cluster. Small insets are snapshots from numerical simulations of one particle trajectory at $B_z = 7.3$mT (left), $B_z = 82.4$mT (center) and $B_z = 210.9$mT (right). **b** Interparticle distance fluctuations $u_{rel}$ as a function of the field amplitude $B_z$ for the confined nanoparticle cluster.

An increase in $B_z$ strengthens the repulsion between the particles and qualitatively changes the density distribution. While one particle still shows the similar Gaussian distribution but increasingly sharper at large $B_z$, for $N = 2, 3$ and $4$ the density distributions $\rho(x, y)$ become significantly nonmonotonic. Indeed, for relatively stronger repulsion, the restoring trap force is no longer able to keep particles distributed uniformly about the central potential well. Instead, they are radially displaced from the trap center and localize at a certain distance from it. Thus, the density field $\rho(x, y)$ shows an angle-independent, ring-like structure, which corresponds to the bimodal distributions $P(\delta x)$ with a local minimum at the center, see Fig. 4c. This picture, however, does not preserve for the further growth in $N$. Starting from $N = 5$, the local minimum at the center flips to a global maximum, while other maxima do not disappear and the distribution slightly broadens. At $N = 5$ and $6$, the clusters contain enough particles to stabilize one particle at the trap center by forming a stable outer ring out of the other particles, which is impossible for smaller $N$. The structure of density plots reflects the shell-like ordering of particles, which is determined by the balance of confining and repulsive forces. Under these forces, the number of local maxima can be understood from the number of rings that can be formed by the given number of particles[34].

**Cluster crystallization and trap escape**. Apart from contrasting the experimental data for a few trapped particles, our numerical mapping model, Eq. (4), allows exploring the assembly and dynamics for large field amplitudes, which are currently unreachable by our experimental setup. These results, however, could be readily tested with other ferromagnetic thin films able to support larger field modulations[35]. Increasing the applied field, leads to a stronger localization of nanoparticles in the trap and simultaneously strengthens the repulsion between them. In a related context, melting of confined few-body systems has been investigated in experiments[36,37] and numerical simulations[38–40] for dipolar particles under hard-wall confinements. However, in contrast to these works, our system is characterized by a nearly harmonic confining potential that can be controlled together with the pairwise interaction.

Figure 5 shows the field-induced crystallization process for a cluster composed of $N = 29$ nanoparticles. We first analyze the structural transition from disorder to order in terms of the time-averaged bond-orientational parameter $\Psi_6$, see "Methods". Generally, its values range from 0 to 1, indicating complete disorder and perfect order, respectively. As shown in Fig. 5a, we can identify three different regimes. For low fields, $B_z < 60$ mT, the particles are thermally delocalized and can explore the whole

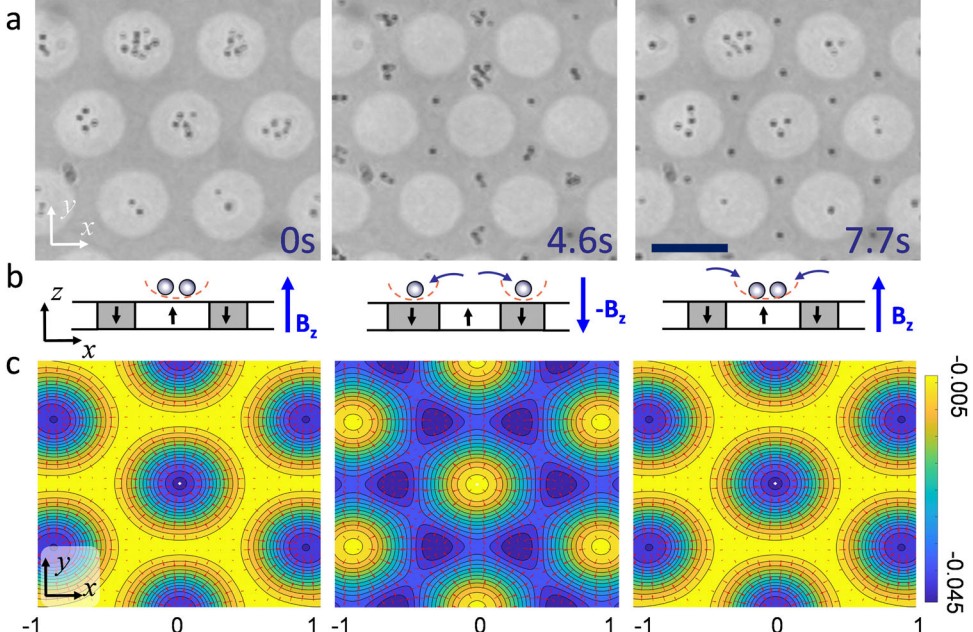

**Fig. 6 Field-induced trap escape. a** Sequence of images showing the nanoparticle exchange process between 7 nearest bubbles. The particle exchange between the magnetic bubbles (white disks) and the interstitial region (gray) is induced by reversing the field at $t = 2.3$ s, $B_z \to -B_z$, and later back to $-B_z \to B_z$ at $t = 4.6$ s. The scale bar is 10 $\mu m$, see also VideoS2 in the Supporting Information. The error bars result from the statistical average of different simulation data. **b** Schematics illustrating the particle positions and the applied field in the $(x, z)$ plane. **c** Energy landscape of the magnetic lattice calculated at an elevation $z = 1.3$ $\mu m$ and for an applied field $B_z = 4.3$ mT (first and third image) and $B_z = -2.7$ mT (second image) as described in "Methods". Energy minima are colored in blue, maxima in yellow. The red arrows superimposed to the images display the magnetic gradient.

bubble area. Increasing $B_z$, the parameter $\Psi_6$ raises rapidly towards a maximum value of $\Psi_6 \approx 0.7$, which is kept stable over a range of $B_z$ from 60 to 150 mT. In this regime of intermediate fields, the nanoparticles start to localize along three concentric shells, with frequent intershell jumping which, however, neither destroys the shell ordering nor suppresses the radial mobility, see insets in Fig. 5a. Finally, for high fields, $B_z > 150$ mT, we observe a decrease in $\Psi_6$ similar to a re-entrant behavior. Now the pairwise interactions are so strong that they impede the intershell jumping, significantly reducing the radial mobility and forcing the particles to stay within the formed concentric shells. In this situation, the angular dynamics within the shell intensifies, while the radial motion freezes, a re-entrant effect that in many cases appears similar to the case of hard-wall confinement of particles[36]. To further quantify the crystallization process we compute another parameter $u_{rel}$, the relative interparticle distance fluctuations[41] (see "Methods"). As shown in Fig. 5b, the relative interparticle distance fluctuations $u_{rel}$ decrease with the increase of the external field, with an amplitude of order similar to that reported for microscopic particles[40]. Further, we observe a clear change in slope of $u_{rel}$ close to the second transition at high field where only intrashell motion occurs.

While the experiments reported until now demonstrate the trapping within the circular Bloch walls, our system allows us also to change the particle location across the lattice and thus to exchange particles between magnetic bubbles. Thus, the trapped nanoparticles are enslaved within the magnetic bubble, and can leave them only by an external command, such as an applied field. This feature is demonstrated in Fig. 6a–c by switching the direction of the applied field $B_z \to -B_z$ thus, the field becomes antiparallel to the bubble magnetization. In this situation, the nanoparticles are forced to jump into the interstitial region (first and second image), while a subsequent field inversion ($-B_z \to B_z$) forces the nanoparticles to return back to one of the nearest bubbles (second and third image), see also Video S2 in the

Supporting Information. As shown in Fig. 6a, the central bubble is initially filled by $N = 5$ particles which are then ejected toward six interstitial places, where they meet other nanoparticles coming from neighboring domains. Thus, each interstitial region receives a fraction of the nanoparticles from the three nearest neighbor bubbles, and after the second field reversal only $N = 3$ have returned back to the original bubble. The exchange process can be understood by analyzing the magnetostatic potential in the particle plane $(x, y)$, see Fig. 6c. When the field is antiparallel to the bubble magnetization, it induces a reversal of the energy landscape and the bubbles correspond not to minima but maxima of the potential, while six potential wells of triangular shape are nucleated around each bubble. This induces a particle displacement at a speed of $v \sim 20$ $\mu m/s$, which is proportional to the local magnetic gradient. As shown in the third image of Fig. 6a, some particles may stay within the interstitial region even after the second field reversal. They are located in an unstable equilibrium, and can easily jump back to a bubble due to thermal fluctuations. Reversing the field polarity may be used to remove nanoparticles from each bubble, which later can be refilled with another field reverse. Thus our virtual magnetic trap may be also used as a rudimentary form of logic memory based on magnetic nanoparticles, which can be stored within the platform by localizing them, and later "erasing" their location by an external command. In particular, a recent work[42] demonstrated the possibility of combining lithographic confinement, electrostatic levitation and external actuation to store and retrieve logic information from levitated nanoparticles. Since our FGF could be easily coupled with a lithographic structure, similar operations could be parallelized on the whole bubble lattice using an external magnetic field.

## Discussion

We demonstrate the controlled magnetic trap to assemble and manipulate clusters of magnetic nanoparticles in solution. We

combine experiments with theory to show that the equilibrium dynamics of the clusters arises from the balance between the confining harmonic potential and repulsive dipolar interactions induced by the substrate and the applied field. The external field serves as the only control knob that may be used to tune both the stiffness of confinement and the strength of interparticle interactions. We further construct a numerical simulation model based on the experimental parameters as a mapping between the experiment and theory, which we exploit to extrapolate the experiment for larger ranges of fields and particle numbers. We investigate the field-induced crystallization process and reveal three different dynamics regimes. Finally, we experimentally demonstrate delocalized transport of nanoparticles across the lattice and the exchange process between the magnetic domains by inverting the applied magnetic field. We note also that trapping of microscale colloids has been achieved both by lithographic wedges[43,44] and high-frequency electric field[45]. However, our work allows us to trap nanoscale magnetic particles via a magnetic landscape with tunable stiffness, keeping them confined to a surface in stable two-dimensional clusters in absence of any topographic relief.

Magnetic nanoparticles are the focus of intense scientific research for their special physical properties which make them widely used in biomedicine[46], microfluidics[47,48], magnetic imaging[49,50] or nanomaterial-based catalysts[51], among others. However, the doping with iron oxide makes these nano-size objects light adsorbing, and thus rather difficult to be reliably trapped via optical tweezers. Although recent progress in this direction has been demonstrated at the single-particle level[52,53], extensions to stable entrapment of ensembles/clusters of nanoparticles in more than one spatial direction remains challenging. With our study, we are able to overcome these restrictions because our setup provides a versatile platform to localize, stably confine and tune the behavior of nanoparticles on a spatially extended surface.

The magnetic bubble lattice can be easily configured by an out of plane magnetic field and the corresponding bubble domains manipulated via a small gradient when the field is still on. Further, given the fast response of the Bloch wall to magnetic field, with a propagation speed of the order of ~ 1km s$^{-1}$ [54], these features make this type of approach a promising candidate for magnetic-based nanoscale trapping. While our magnetic domains are relatively large, much smaller bubbles could be also synthesized in garnet film[55] that could eventually lead to further system miniaturization. Moreover, the proposed technique can also be extended to biological systems such as magnetotactic bacteria[56] and different dispersing media such as viscoelastic fluids[57]. In the latter case, plans to use our trapped nanoparticles as strongly thermal microrheological probes are currently under research.

## Methods

### Experimental system

*Preparation of the ferrite garnet film (FGF).* We coat the FGF with a thin polymer film (1 μm) made of a positive photoresist AZ-1512 (Microchem Newton, MA). The coating increases the effective particle elevation from the FGF, which has a twofold effect. First, it makes the potential effectively parabolic. Second, it reduces the otherwise strong attraction by the Bloch walls, preventing the particle sticking to the FGF. The photoresist is applied via spin coating (Spinner Ws-650Sz, Laurell) and subsequent UV photo-crosslinking (Mask Aligner MJB4, SUSS Microtec). We use the following procedure. First, we clean the FGF in an ultrasonic bath filled with acetone (Merck) for ~15 min and then with isopropanol (Sigma) for the same amount of time. After that, the FGF is dried under a stream of N₂. Few drops of the photoresist are deposited above the cleaned FGF, and then dispersed using a spin coater working at 3000 rpm for 30 s. After that, the polymer is baked by placing FGF above a plate heated at 95 °C for 1 min, and then photo-crosslinked via exposure to 5 s of UV irradiation at a power of $P = 30$ mW/cm². A final post-

bake process is applied by placing the FGF above a hotplate heated at 115 °C for 50 s. The latter process induces further hardening of the photoresist above the FGF.

*Preparation of colloidal solution.* We dilute a drop of the stock solution of the nanoparticles in highly deionized water (MilliQ, Millipore). The particles become electrostatically stabilized in water by the negative charges acquired from the dissociation of the surface carboxylic groups (COOH). We tune the amount of H₂O to reach a density of ~10⁷ beads/ml and add few drops above the magnetic film. After a few minutes, the particles sediment to the FGF surface due to density mismatch and they remain confined above it without sticking. We wait for 15 min to allow the sedimentation of all particles, place above a coverslip (no.1, Thermo Scientific Menzel) and use an immersion oil (Immerso 1111-806, Zeiss) between the coverslip and the microscope objective (100 × 1.3NA, Nikon).

*Magnetic field and particle tracking.* The external magnetic field perpendicular to the FGF is applied by using a custom-made coil placed below the magnetic film (see Fig. 1a). The coil is composed of 700 turns of a 0.5 mm thick copper wire and is connected to a DC power supplier (TTi El 302), which allows to apply a spatially uniform, static magnetic field up to $B_z \simeq 20$ mT. We use a teslameter (FM 205, Projekt Elektronik GmbH) to calibrate the field amplitude and determine the homogeneity of the field distribution around the sample plane. We find that for the amplitudes used ($B_z \leq 6$ mT) the field is spatially uniform above the sample area of ~1 cm². The particle positions are determined via digital video microscopy[58]. We use an upright optical microscope (Eclipse Ni, Nikon) equipped with a 100 × 1.3 NA oil immersion objective and a CCD camera (Basler Scout scA640-74fc, Basler) to record experimental movies at 75 frames per second.

### Theoretical model

#### Derivation of the model

*General framework.* The experimental measurements are interpreted within a model that describes overdamped motion of an ensemble of $i = 1, ..., N$ nanoparticles with positions $\mathbf{r}_i = (x_i, y_i)$ in a potential $U$ and subject to thermal fluctuations,

$$\zeta \frac{d\mathbf{r}_i(t)}{dt} = -\frac{\partial}{\partial \mathbf{r}_i} U(\mathbf{r}_i, \mathbf{r}_j) + \boldsymbol{\xi}_i(t) , \qquad (5)$$

where $\zeta$ is the friction coefficient, $\boldsymbol{\xi}$ is the Gaussian white noise with zero mean and diagonal covariance of strength $2\zeta k_B T$ with $k_B T$ being the thermal energy. The potential $U(\mathbf{r}_i, \mathbf{r}_j) = U_1(\mathbf{r}_i) + \sum_{j>i} U_2(\mathbf{r}_i, \mathbf{r}_j)$, comprising the single-particle interaction with the external field, $U_1$, and pairwise interactions with all other particles, $U_2$.

A spherical paramagnetic particle of diameter $d$ in a magnetic field $\mathbf{H}$ behaves as an induced magnetic moment $\mathbf{m} = \upsilon \chi \mathbf{H}$, where $\chi$ is the effective magnetic susceptibility and $\upsilon = \pi d^3/6$ is the volume of particle. Therefore, considering $U_1(\mathbf{r}_i) = -(\mathbf{m}_i \cdot \mathbf{B}_i)/2$ with $\mathbf{B}_i = \mathbf{B}(\mathbf{r}_i)$ as the energy of the dipole-field interaction and $U_2$ as the energy of dipole-dipole interactions, we have

$$U_1(\mathbf{r}_i) = -\frac{\upsilon \chi \mathbf{B}^2(\mathbf{r}_i)}{2\mu_0} , \qquad (6)$$

$$U_2(\mathbf{r}_i, \mathbf{r}_j) = \frac{(\mathbf{B}_i \cdot \mathbf{B}_j) - 3(\hat{\mathbf{r}}_{ij} \cdot \mathbf{B}_i)(\hat{\mathbf{r}}_{ij} \cdot \mathbf{B}_j)}{4\pi \mu_0 r_{ij}^3} , \qquad (7)$$

where $\mathbf{r}_{ij} = \mathbf{r}_i - \mathbf{r}_j$, $\hat{\mathbf{r}}_{ij} = \mathbf{r}_{ij}/r_{ij}$ and $r_{ij} = |\mathbf{r}_{ij}|$. Note that because the solvent is nonmagnetic, the magnetic induction $\mathbf{B} = \mu_0 \mathbf{H}$ with $\mu_0 = 4\pi \times 10^{-7}$ H m$^{-1}$ the magnetic permeability of free space.

*Exact stray field above the FGF.* The stray field generated at the surface of the magnetic bubble lattice, $\mathbf{B}_{\text{stray}}$, can be calculated exactly by summing up the field from a triangular lattice with period $a$ of magnetic bubbles with lattice vectors $\mathbf{p} = n\mathbf{a}_- + m\mathbf{a}_+$ with $\mathbf{a}_\pm := (\sqrt{3}a/2, \pm a/2)$, and $n, m$ integers. Each bubble is considered as a cylindrical uniformly magnetized ferromagnetic domain, generating a stray field above its surface written in cylindrical $(r, z)$ coordinates as[59]: $\mathbf{b} = \hat{\mathbf{r}} b_r(r, z, t) + \hat{\mathbf{z}} b_z(r, z, t)$ with

$$b_r = \frac{B_s}{\pi} \sqrt{\frac{D}{2r}} Q_{\frac{1}{2}}\left(\frac{r^2 + D^2/4 + z^2}{rD}\right), \qquad (8a)$$

$$b_z = B_s - \frac{B_s}{\pi} \left[\kappa_- \Pi(n_+|K) + \kappa_+ \Pi(n_-|K)\right]. \qquad (8b)$$

Here, $B_s$ is the saturation magnetization of the FGF, $\mathbf{r} = (x - p_x)\hat{\mathbf{x}} + (y - p_y)\hat{\mathbf{y}}$, $Q_n$ is the Legendre function of the second kind and order $n$, $\Pi(n, m)$ gives the complete elliptic integral of the third kind, $K = \sqrt{2rD/[z^2 + (r + D/2)^2]}$; $n_\pm = 2r/(r \pm \sqrt{r^2 + z^2})$, the bubble diameter depends on the external field $\mathbf{B}_{\text{ext}}$

and can be expressed as: $D(t) = 2a\sqrt{(\hat{\mathbf{z}} \cdot \mathbf{B}_{ext}(t)/B_s + 1)\sin(\pi/3)/2\pi}$ and $\kappa_{\pm} = (\sqrt{r^2 + z^2} \pm D/2)(\sqrt{r^2 + z^2} \pm r)/[z\sqrt{(r + D/2)^2 + z^2}]$.

The field generated by such array of bubbles is given by $\mathbf{B}_b = \sum_{n,m} \mathbf{b}_{nm}$ with the indexes $n$ and $m$ over the entire triangular lattice. The overall substrate field is obtained as $\mathbf{B}_{stray} := \mathbf{B}_b - \mathbf{B}_f$, where $\mathbf{B}_f$ is the contribution due to the oppositely magnetized film calculated for a cylindrical domain covering the entire sample area. This approach is used to calculate the landscapes in Figs. 1e and 6c.

*Reduced description.* The total magnetic field above the substrate $\mathbf{B} = \mathbf{B}_{ext} + \mathbf{B}_{stray}$, where the external field $\mathbf{B}_{ext} = B_z \hat{\mathbf{z}}$ and an approximate solution for the stray field that captures triangular symmetry of the magnetic bubble lattice (see supplementary information in Ref. [60]) can be represented as $\mathbf{B}_{stray}(x, y, z) = B_s e^{-\kappa z}$ $\mathbf{h}(x, y)$, $\mathbf{h} = (h_x, h_y, h_z)$ with $h_x = \sin\kappa x + \sin(\kappa x/2)\cos(\sqrt{3}\kappa y/2)$, $h_y = \sqrt{3}\cos(\kappa x/2)\sin(\sqrt{3}\kappa y/2)$, $h_z = \cos\kappa x + 2\cos(\kappa x/2)\cos(\sqrt{3}\kappa y/2)$. Here, $z$ is the particle elevation above the substrate, and $\kappa = 4\pi/(\sqrt{3}a)$ with the lattice constant $a$. Expanding the solution about the center of a magnetic bubble, $\kappa r \ll 1$, we arrive at the reduced analytic expression

$$\mathbf{B} = \frac{3}{4} B_s e^{-\kappa z} (2\kappa x, 2\kappa y, 4 - \kappa^2 r^2) \qquad (9)$$

with $r^2 = x^2 + y^2$. Employing this result in Eq. (6), we obtain a harmonic approximation for the confining potential, $U_1(r) = k_e r^2/2$ (Eq. (1)), with a field-dependent spring constant of the form, $k_e(B_z) = k_0(1 + c_1 B_z)$, where $k_0$ and $c_1$ are fitting constants. To evaluate the pairwise interaction potential, Eq. (7), we stick to the limit $\kappa r \to 0$ in Eq. (9), which reveals central strictly repulsive interactions between the particles, $U_2(r) = \gamma/r^3$ (Eq. (3)), with the field-depenent repulsion strength $\gamma(B_z) = \gamma_0(1 + c_2 B_z)^2$ and fitting constants $\gamma_0$ and $c_2$. Accounting for these results in Eq. (5), we obtain the governing equation, see Eq. (4), serving as a mapping between the theory and experiment.

### Fitting the model to the experiment

*Particle trapping.* Properties of thermal motion of individual particles confined to the harmonic potential $U_1(r) = k_e r^2/2$ are drawn from the single-particle stationary probability distribution $P_1(\mathbf{r}) \propto \exp[-\beta U_1(\mathbf{r})]$, where $\beta^{-1} = k_B T$. The effective spring constant $k_e(B_z) = k_0(1 + c_1 B_z)$ is evaluated from the fits of the form $k_e r^2/2$ against the data represented as $-\ln[P_1(r)/P_1(0)]$. The constant $\beta k_0 \approx 3.174 \, \mu m^{-2}$ (which corresponds to $k_0 \approx 0.0129 \, pN/\mu m$) follows from the case $B_z = 0 \, mT$ and the value $c_1 \approx 0.333 \, \mu T^{-1}$ is recovered by satisfying the measured profiles of potential at different $B_z$, see Fig. 2b, c.

Alternatively, the values of $k_e$ can be extracted from the mean-square displacement (MSD), $\langle\Delta r^2\rangle(t) = \langle(\mathbf{r}(t + t') - \mathbf{r}(t'))^2\rangle$, with a lag time $t$. As follows from Eq. (4) at $\gamma = 0$, we obtain a Langevin equation describing Ornstein-Uhlenbeck process, $\zeta\dot{\mathbf{r}}(t) = -k_e \mathbf{r} + \boldsymbol{\xi}(t)$. The motions in $x$- and $y$-directions decouple and remain similar, leading to[33] $\langle\Delta x^2\rangle(t) = \langle\Delta y^2\rangle(t) = (\beta k_e)^{-1}[1 - \exp(-t/\tau)]$ with $\tau = \zeta/(2k_e)$. Therefore, the full MSD is evaluated as $\langle\Delta r^2\rangle = \langle\Delta x^2\rangle + \langle\Delta y^2\rangle$, to yield Eq. (2), with the long-time limit $\langle\Delta r^2\rangle \to 2k_B T/k_e$, as $t \to \infty$. The time-dependent MSDs with the spring constant evaluated as described next are shown in Fig. 2d. Note that $\tau = \tau(B_z)$ with $k_e/\zeta$ represented as $\beta k_e D_0$, where $D_0 = (\beta\zeta)^{-1} \approx 1.04 \, \mu m^2/s$ is the free diffusion coefficient.

The originally anharmonic nature of confining potential may lead to deviations in effective spring constants measured from the particle distribution $P_1(r)$ and from the MSD. For the latter, our measurements indeed show slightly smaller values of $k_e$, a known effect that is well captured by small anharmonic corrections to the potential, see Appendix B[61]; for limited statistics, they may not be well detectable from measuring $P_1(r)$ because of its sharp exponential form. Within our model, they are equivalent to a coordinate-dependent spring constant $k_e'(r) = k_e(1 - c_4 r^2/4)$. Evaluating its average, we remain within the harmonic approximation for $U_1$ with a slightly modified spring constant $\langle k_e'\rangle = \int k_e'(r) P_1(\mathbf{r}) d\mathbf{r} = k_e - \Delta k_e$, $\Delta k_e = c_4 k_B T/2 > 0$, in quantitative accord with our MSD data. Because the constant correction $\Delta k_e$ is otherwise irrelevant for our study, we do not take it into account.

*Pairwise interactions.* The parameters of pairwise repulsion interaction are evaluated from the case of two particles, for which the total potential $U(\mathbf{r}_1, \mathbf{r}_2) = U_1(r_1) + U_1(r_2) + U_2(r)$. Here, $r = |\mathbf{r}_1 - \mathbf{r}_2|$ is the interparticle distance. The corresponding two-particle stationary probability distribution $P_2(\mathbf{r}_1, \mathbf{r}_2) \propto \exp[-\beta U(\mathbf{r}_1, \mathbf{r}_2)]$ can be projected to $P_2(r) = \int P_2(\mathbf{r}_1, \mathbf{r}_2)\delta(|\mathbf{r}_1 - \mathbf{r}_2| - r)d\mathbf{r}_1 d\mathbf{r}_2 \propto \exp[-\beta U(r)]$. Therefore, we evaluate $U(r)$ as $-\ln[P_2(r)/P_{max}]$ by measuring $P_2(r)$, where $P_{max} = \max_r P_2(r)$; with this normalization $U_{min} = \min_r U(r) = 0$. We match the corresponding data from the experiment and simulations, see Fig. 3, by keeping the trap parameters as described above and tuning the strength $\gamma(B_z) = \gamma_0(1 + c_2 B_z)^2$ of repulsive interactions, $U_2(r) = \gamma/r^3$. We find the best fit for $\beta\gamma_0 \approx 2.4 \, \mu m^3$ for $B_z = 0 \, mT$ and $c_2 = 0.07 \, mT^{-1}$, which accounts for the field dependence.

For a system with an elongated, quasi-one-dimensional trap, for which the most probable configuration corresponds to $\mathbf{r}_1 = -\mathbf{r}_2 = (r/2, 0)$ with the trap center at origin, and a central pairwise potential $U_2(r)$ a reliable analytic approximation is

available[62], $U(r) \approx 2U_1(r/2) + U_2(r)$. For our two-dimensional trap, this estimate is still valid, because due to axial symmetry, cf. Fig. 4b, the most probable configuration can be similarly represented as $\mathbf{r}_1 = -\mathbf{r}_2 = \mathbf{r}/2$, leading to the approximation $P_2(r) \propto \exp[-\beta(2U_1(r/2) + U_2(r))]$. The relative weights of other possible configurations seem weak to introduce significant errors, and the approximate formula $U(r) \approx 2U_1(r/2) + U_2(r)$ remains a robust reference also in two dimensions, see Fig. 3a.

**Analysis of collective states.** To study collective states, cluster crystallization and trap escape, we intergrate Eq. (4) numerically. The parameters of the trap and repulsion, including their field dependence, are taken as determined in the single- and two-particle setups. We have additionally ensured that every particle interacts with all other particles. This way, we have access to particle positions within the numerical model. Experimentally, particle positions are extracted by particle tracking. These data are used to evaluate and compare such quantities as mean-square displacement and particle distributions.

To quantify the degree of local hexagonal ordering, from particle positions we also calculate the bond-orientational order parameter, defined as $\psi_{6,k} = N_b^{-1}|\sum_{j=1}^{N_b} \exp(6i\theta_{kj})|$, where $N_b$ is the number of neighbors of particle $k$, and $\theta_{kj}$ is the angle between a fixed axis and the bond joining particles $k$ and $j$. Further we average it at each time step over all particles. Following a previous work on optically trapped colloidal cluster[63], to minimize artefacts from the curved frontier we calculate $\psi_6(t) = \langle\psi_{6,k}\rangle$, by considering only the particles not adjacent to the outer boundary. Finally, we perform a running-time average and obtain the measure $\Psi_6 = \overline{\psi_6(t)}$ for an entire simulation, as shown in Fig. 5. Another order parameter that we use to quantify the crystallization process is the relative interparticle distance fluctuations[41], $u_{rel} = 2/(N(N-1))\sum_{1 \le i<j}^N [\langle r_{ij}^2\rangle/\langle r_{ij}\rangle^2 - 1]^{1/2}$, where $r_{ij}$ is the distance between particles $i$ and $j$, $N$ is the total number of particles, and we have also performed the running-time averaging.

## Data availability

The data that support the findings in this study are available within the article and its Supplementary information. Further details are available from the corresponding authors upon reasonable request.

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

## Acknowledgements

This work has received funding from the European Research Council (ERC) under the European Union'sHorizon 2020 research and innovation programme (grant agreement No 811234). P.T. acknowledges support from the Spanish MINECO (FIS2016-78507-C2-2-P, ERC2018-092827) and from the Generalitat de Catalunya under program "Icrea Academia". A.V.S. acknowledges support by Deutsche Forschungsgemeinschaft (DFG) through grant SFB 1114, project no. 235221301, sub-project C03.

## Author contributions

P.T. and A.V.S. conceived the project and wrote the manuscript. P.T. designed the experiment and performed the measurements, T.H. synthesized the FGF, A.V.S. developed the theoretical model and performed simulations. All authors discussed the results and commented on the manuscript at all stages.

## Competing interests

The authors declare no competing interests.
