## [Peer Review File · Nature Communications]

Reviewers' Comments:

Reviewer #1:

Remarks to the Author:

In this manuscript Tierno et al. developed a technique to generate an array of magnetic traps to assemble and manipulate clusters of magnetic nanoparticles in solution, and to study in details the interactions among them that control the formation of collective states. Providing methods and tools for the stable entrapment of ensembles of nanoparticles in more than one spatial direction is a key step to study adsorption, mobility, emerging collective phenomena, with a potential impact in many research fields, like biomedicine, microfluidics, magnetic imaging, nanoscale friction, nanomaterial-based catalysis to mention some.

The platform developed here is based on the generation of a triangular lattice of micron-size magnetic bubble domains, of cylindrical shape, in a uniaxial ferrite garnet film (with perpendicular anisotropy) epitaxially grown on a GGG (gadolinium gallium garnet) substrate. The triangular lattice of magnetic bubbles is induced by subjecting the film to an alternating magnetic field. The application of a vertical magnetic field provides a knob to further control the diameter of the bubble. The basic idea is not new since the authors have proposed and utilized it in the past [Ref. 30 of the manuscript and Phys. Rev. Lett. 99,038303 (2007)] to study the motion of colloidal particles on a magnetic bubble lattice. In any case, I consider the research presented here a clear physics and methodology advance with respect to the previously published work. The key point in the present work is that the magnetostatic potential generated (and actively controlled by a magnetic field) by the bubble is largely harmonic and this allows for the construction of a robust simulation model, which in turn enables the precise extraction of parameters (mobility, pairwise interaction, effective spring constant) and a detailed study of nanoparticle dynamics.

The quality of the experiments and the depth of the data analysis are outstanding. Therefore, I am prone to recommend the manuscript for publication.

However, there are several minor points that should be addressed prior to publication.

- I am puzzled by the following question: the potential landscape shown in Fig. 3e is very different from the one shown in Ref. 30 although the authors refer to that work for its calculation, given that the material and the magnetic bubble lattice is the same (or similar). In Ref. 30 the potential (in zero or nearly zero applied perpendicular magnetic field) shows maxima on the bubbles and minima around them. Indeed, magnetic nanoparticles were found to move around the bubbles [PRL 99,038303 (2007)] and not assembling on the bubbles. In the present work, it is the opposite. Furthermore, the potential shown in Ref. 30 is far from being harmonic. What is that I am missing here?

In a recent study where single-crystal cylindrical nanowires of Fe, Co, and Ni, also arranged on a triangular lattice, were synthesized in Al₂O₃ membrane, the magnetic stray field in the z-direction as well as the magnetostatic potential, was found to be the strongest at the edge of the wire [Y. P. Ivanov et al., ACS applied materials & interfaces 11, 4678-4685 (2019)]. This is expected considering simple magnetostatic concepts. The reason for having a different situation here seems to be related to the fact that the cylindrical bubbles are embedded in an opposite magnetized environment. Maybe the authors might comment and discuss a bit this point.

- Related to the previous point, in the present manuscript, a certain emphasis is put on the existence of magnetic Bloch walls separating the bubbles from the background with opposite magnetization. However, this point is not developed further and the calculations in Ref. 30, referred to in the present manuscript, seem not considering the presence of such walls (at least there is no reference thereto such walls). In any case, the Bloch walls should have been present also in the study reported in PRL 99,038303 (2007), thereby I still cannot catch the reason for having such different magnetostatic potential landscapes in the two studies.

- Still related to the previous points: how the potential landscape changes with elevation? The choice of capping the garnet film with 1 micron-thick polymer layer is random or the result of a systematic study?

- A reference (or a justification) for the choice of the value (approx. 2) of the effective magnetic susceptibility should be provided.

- Page 5, left column, "long-time spatial density distribution" should be "stationary spatial density distribution" as reported in the Fig. 4 caption.

- Is the scale bar corresponding to 2 μm shown in Fig. 4b correct? The plots of $P(\Delta_x)$ shown in Fig. 4c for $B_z = 4.7$ mT look slightly wider than the 2D density distributions shown in Fig. 4b...but

it could be a visual impression due to the color scale used.

- I understand that results shown in Fig. 5 come from simulations; however, I wonder what is the utility of the study where the field B_z extends to 250 mT, considering that the material saturates at a much weaker B_z (an H_c of 11.4 mT is reported in page 2), so that no bubble would be present above 10-15 mT?

Reviewer #2:

Remarks to the Author:

This paper reports experimental observations of colloidal particles on a 2D array of ferromagnetic domains. The authors use theory and simulation to replicate single and multi-particle distributions. Results include a model for particle interactions with the applied field and dipoles of nearby particles. Fluid clusters are characterized based on order parameters and fluctuations. It is suggested that some of the clusters are crystalline, although this is not obvious based on established 2D freezing criteria. Single particle diffusion within energy wells is characterized and escape of particles from wells is also documented. The results provide an interesting demonstration of colloidal clustering on magnetic patterns with quantification of dipolar interactions and some dynamic features. Some aspects of the analysis and findings are similar to prior studies of colloids on 2D energy landscapes mediated by gravity^{1,2} or dipolar potentials.³ The novelty and significance of findings in the present study could perhaps be better demonstrated through comparison and contrast with these studies and other relevant precedent. The paper is well written and organized and the findings are generally supported by the results and their discussion.

1. Bahukudumbi, P. and M.A. Bevan, Imaging Energy Landscapes using Concentrated Diffusing Colloidal Probes. *J. Chem. Phys.*, 2007. 126: p. 244702.
2. Fernandes, G.E., D.J. Beltran-Villegas, and M.A. Bevan, Spatially Controlled Reversible Colloidal Self-Assembly. *J. Chem. Phys.*, 2009. 131: p. 134705.
3. Juarez, J.J. and M.A. Bevan, Interactions and Microstructures in Electric Field Mediated Colloidal Assembly. *J. Chem. Phys.*, 2009. 131: p. 134704.

Reviewer #3:

Remarks to the Author:

The authors present a nice study on magnetic trapping of nanoparticles in a magnetic field landscape. The study is clean, thorough and well executed. The experiments describe interesting regimes of tunable collective magnetic nanoparticle behaviour with the interparticle interaction strength tuned using the external magnetic field.

While the system has some degree of tunability in that the interaction strengths can be tuned using an external B field, the general properties of the landscape still seem hardwired into the system by assembly of the domain walls. Can the underlying domain wall structure be reconfigured at will - if so, could the authors comment on how and on what timescales? This would be an important feature to highlight that contrast the approach to nanofabrication-based trapping techniques where the landscape is largely hardwired by lithography as the authors mention. Also, what is the smallest effective domain size achievable for this type of system? This could be important to know as it will ultimately limit the maximum particle density in any possible high density array type of application.

A few minor points:

It is not clear very clear what the term "enslaved" in the title really means. Do the authors possibly mean something along the lines of "confined"? The term is not mentioned or defined anywhere in the manuscript. If this is field specific jargon the suggestion is to use a different word/more common technical term or defining the term in the text for a broad interdisciplinary audience.

Towards the end of the manuscript the authors refer to the possibility of using their approach for storing and erasing information. Do the authors see any prospects for realising a magnetic colloid-based memory/data storage system along the lines of the electrostatic version described, e.g., in Myers et al., *Nat. Nano.* 10 (2015)? Some comments in this direction may help to flesh out possible technological applications of the technique.

Response to Reviewers' comments

We thank the Referees for their thorough reading of our manuscript and for the constructive suggestions/criticisms. Our point-by-point responses and the corresponding changes made to the manuscript are described below.

Response to Reviewers #1

Referee: *In this manuscript In any case, I consider the research presented here a clear physics and methodology advance with respect to the previously published work. The key point in the present work is that the magnetostatic potential generated (and actively controlled by a magnetic field) by the bubble is largely harmonic and this allows for the construction of a robust simulation model, which in turn enables the precise extraction of parameters (mobility, pairwise interaction, effective spring constant) and a detailed study of nanoparticle dynamics. The quality of the experiments and the depth of the data analysis are outstanding. Therefore, I am prone to recommend the manuscript for publication.*

Authors: We thank the Referee for all the positive comments on our manuscript. We also thank the Referee for recognizing the “*quality of the experiments and the depth of the data analysis*”.

Referee: *However, there are several minor points that should be addressed prior to publication.*

- I am puzzled by the following question: the potential landscape shown in Fig. 3e is very different from the one shown in Ref. 30 although the authors refer to that work for its calculation, given that the material and the magnetic bubble lattice is the same (or similar). In Ref. 30 the potential (in zero or nearly zero applied perpendicular magnetic field) shows maxima on the bubbles and minima around them. Indeed, magnetic nanoparticles were found to move around the bubbles [PRL 99,038303 (2007)] and not assembling on the bubbles. In the present work, it is the opposite. Furthermore, the potential shown in Ref. 30 is far from being harmonic. What is that I am missing here?

Authors: We thank the Referee for raising this important point, which is twofold:

i) **Physically different conditions of observation**. Effectively in Ref. 30 and in the other mentioned work [PRL 99,038303 (2007)], the paramagnetic colloids were observed to circulate around the bubble, thus localized at the interstitial regions, rather than being trapped in the centre of the bubble. However, there is an important difference between those two works and our current manuscript. In the previous works, the FGF film was subjected to a precessing magnetic field characterized by a static, out of plane component which was applied perpendicular to the film and antiparallel to the bubble magnetization, please see Fig.1(a) of [PRL 99,038303 (2007)]. This causes the bubbles to shrink, enlarging the interstitial region, and creates energy minima at the bubble perimeter, as shown also in Fig.6 of the current paper for a magnetic field applied antiparallel to the bubble magnetization. Under such condition, the effective magnetostatic potential acquires a complex shape, and it is indeed non-harmonic, as correctly noticed by the Referee and as can be seen from Fig.6(c), central panel. In contrast, the main focus of our work is the opposite situation when the field is applied parallel to the bubble magnetization direction. In this case, the bubbles enlarge their shape, and the energy minima are located at the centres of the

cylindrical domains (see Fig.6(c) left or right panel), where to a large extent the potential remains harmonic.

ii) Technical point on evaluation of the magnetostatic potential. Further, related to this question, and the other two (third and fourth) issues below raised by the Referee, we should apologize for a slightly inaccurate statement about the calculation of the energy landscape showed in Fig.1(e). We stated that we calculated the landscape using the code developed in Ref. 30, which is based on the direct integration of the Bessel functions. Although it eventually leads to the same result for the stray field of the FGF, it appears to be cumbersome. Instead, we have made use of an alternative, but numerically more efficient representation suggested in appendix A of [Y. S. Lin, P. J. Grundy, *J. Appl. Phys.* 45, 4084, 1974] combined with the principle of superposition.

Thus, to exclude any potential confusion for the reader, we have added details of calculations of the field used in our work in Figs. 1 and 6, see the Method section of the revised manuscript.

Referee: *In a recent study where single-crystal cylindrical nanowires of Fe, Co, and Ni, also arranged on a triangular lattice, were synthesized in Al₂O₃ membrane, the magnetic stray field in the z-direction as well as the magnetostatic potential, was found to be the strongest at the edge of the wire [Y. P. Ivanov et al., *ACS applied materials & interfaces* 11, 4678-4685 (2019)]. This is expected considering simple magnetostatic concepts. The reason for having a different situation here seems to be related to the fact that the cylindrical bubbles are embedded in an oppositely magnetized environment. Maybe the authors might comment and discuss a bit this point.*

Authors: We thank the Referee for bringing to our attention this interesting work and the closely related observation, which now we have cited in the main text. Effectively, as the Referee correctly points out, the main difference between our bubble lattice and the regular array of nanowires in this reference is the presence of the oppositely magnetized “ambient” that impedes the formation of vortex stray field along the Bloch walls. We comment this point in the text by writing, on page 2, column 1:

“We also note that recently a strong edge stray field has been generated by an array of regular nanorod assembled in a triangular lattice [New Reference]. However, the presence of the oppositely magnetized surrounding impedes the formation of localized vortices in our system.”

*[New Reference] Y. P. Ivanov, J. Leliaert, A. Crespo, M. Pancaldi, C. Tollan, J. Kosel, A. Chuvilin, P. Vavassori, *ACS Appl. Mater. Interfaces*, 11, 4678–4685 (2019)*

Referee: *- Related to the previous point, in the present manuscript, a certain emphasis is put on the existence of magnetic Bloch walls separating the bubbles from the background with opposite magnetization. However, this point is not developed further and the calculations in Ref. 30, referred to in the present manuscript, seem not considering the presence of such walls (at least there is no reference thereto such walls). In any case, the Bloch walls should have been present also in the study reported in *PRL* 99,038303 (2007), thereby I still cannot catch the reason for having such different magnetostatic potential landscapes in the two studies.*

Authors: Please see our answer above as to why the stray fields are so different for the two cases. Briefly, the shape of the potential strongly depends on the orientation (antiparallel vs parallel) of the applied field with respect to the bubble magnetization direction. Regarding the presence of the

Bloch wall, we confirm that our models do account for their presence. However, we are interested in the stray field generated by the magnetic lattice only at the surface of FGF (or, more generally, above the FGF). Therefore, field calculations (see the updated Method section of the manuscript) typically base on the assumption of instant change of magnetization orientation in the bulk of FGF while crossing the Bloch wall, which is accurate enough to reliably capture all our observations.

Referee: - *Still related to the previous points: how the potential landscape changes with elevation? The choice of capping the garnet film with 1 micron-thick polymer layer is random or the result of a systematic study?*

Authors: The Referee spots an important issue. The exact dependence with the elevation (z) is complicated. Approximate models indicate that the leading contribution to this dependence shows exponential decay with z , i.e. typically very quickly.

To be more precise and specific, we have used our program to demonstrate how the energy landscape changes by varying the elevation of the magnetic particle. The results are shown in the underlying image, where we illustrated different cuts of the energy landscape generated by the stray field of the FGF along a diagonal direction. The locations of the Bloch walls are indicated by black dashed lines, while all lengths are normalized with respect to the lattice constant $a = 11.8$ microns. As can be seen, the stray field indeed changes dramatically when raising the elevation. At the particle elevation of order 1 micron, see the curve for $z = 1.3$ micron ($z/a = 0.11$), and higher the potential can be effectively approximated by a parabola (see red line being fit through the data). Note that (compare to low elevations where particles are attracted to Bloch walls) even the presence of Bloch walls does not break down the harmonic approximation.

Regarding the choice of covering the FGF film with a thin polymer, the reason is twofold. First, as shown above, the potential becomes essentially parabolic, and the particles can be stably trapped inside the magnetic bubbles. Second, the strong magnetic attraction towards the Bloch wall could easily force the particles to stick to the surface and immobilize them there. However, capping the FGF with the polymer coating allows us to weaken the attraction (recall the drastic dependence on the elevation) and avoid sticking to the surface of the FGF. We have commented this point in the text by writing:

“The coating increases the effective particle elevation from the FGF, which has a twofold effect. First, it makes the potential effectively parabolic. Second, it reduces the otherwise strong attraction by the Bloch walls, preventing the particle sticking to the FGF.”

Referee: - A reference (or a justification) for the choice of the value (approx. 2) of the effective magnetic susceptibility should be provided.

Authors: We have added the corresponding reference (now Ref. 31) in the main text.

Referee: - Page 5, left column, “long-time spatial density distribution” should be “stationary spatial density distribution” as reported in the Fig. 4 caption.

Authors: We thank the Referee for noticing this imprecision. We have corrected the corresponding phrase in the text.

Referee: - Is the scale bar corresponding to 2 μm shown in Fig. 4b correct? The plots of $P(\Delta x)$ shown in Fig. 4c for $B_z = 4.7 \text{ mT}$ look slightly wider than the 2D density distributions shown in Fig. 4b...but it could be a visual impression due to the color scale used.

Authors: We thank the Referee for spotting this apparent inconsistency. Effectively, the scale bar shown in Fig. 4(b) is for 3 and not 2 microns, thus slightly larger (the scale bar in Fig. 4(a) is still correct). We thus have adjusted the scale bar in the image to be consistent with that of Fig. 4(a).

Referee: - I understand that results shown in Fig. 5 come from simulations; however, I wonder what is the utility of the study where the field B_z extends to 250 mT, considering that the material saturates at a much weaker B_z (an H_c of 11.4 mT is reported in page 2), so that no bubble would be present above 10-15 mT?.

Authors: Indeed, in Fig.5 we have explored the crystallization scenario of the assembled nanoparticles for very large field amplitude. It is true that above a field of $\sim 15\text{mT}$ in this particular FGF the magnetic bubble disappears. However, there are many other bubble-like ferromagnetic films that are characterized by frozen ferromagnetic domains that can support much higher applied field. An example is represented by cobalt-based magnetic film lithographically patterned via ion bombardment, see [J. Loehr et al. *Commun. Phys.* 1, 4 (2018)]. Further, since the particles are paramagnetic, increasing the field raises the induced moment and thus the dipolar interactions between them. Effectively, the same effect can be obtained by using nanoparticles with higher magnetic susceptibility, namely values of χ that would require smaller field amplitude. We comment this point in the text, and write:

“These results, however, could be readily tested with other ferromagnetic thin films able to support larger field modulations [New Reference 1].”

[New Reference 1] J. Loehr et al. Commun. Phys. 1, 4 (2018).

Response to Reviewers #2

Referee: *This paper reports experimental observations of colloidal particles on a 2D array of ferromagnetic domains. The authors use theory and simulation to replicate single and multi-particle distributions. Results include a model for particle interactions with the applied field and dipoles of nearby particles. Fluid clusters are characterized based on order parameters and fluctuations. It is suggested that some of the clusters are crystalline, although this is not obvious based on established 2D freezing criteria. Single particle diffusion within energy wells is characterized and escape of particles from wells is also documented. The results provide an interesting demonstration of colloidal clustering on magnetic patterns with quantification of dipolar interactions and some dynamic features.*

Authors: We thank the Referee for the positive comments on our manuscript.

Referee: *Some aspects of the analysis and findings are similar to prior studies of colloids on 2D energy landscapes mediated by gravity^{1,2} or dipolar potentials.³ The novelty and significance of findings in the present study could perhaps be better demonstrated through comparison and contrast with these studies and other relevant precedent.*

1. Bahukudumbi, P. and M.A. Bevan, Imaging Energy Landscapes using Concentrated Diffusing Colloidal Probes. J. Chem. Phys., 2007. 126: p. 244702.

2. Fernandes, G.E., D.J. Beltran-Villegas, and M.A. Bevan, Spatially Controlled Reversible Colloidal Self-Assembly. J. Chem. Phys., 2009. 131: p. 134705.

3. Juarez, J.J. and M.A. Bevan, Interactions and Microstructures in Electric Field Mediated Colloidal Assembly. J. Chem. Phys., 2009. 131: p. 134704.

Authors: We thank the Referee for bringing to our attention these very nice articles, which we find relevant and now cite in our manuscript. The first two works demonstrate the confinement of microscopic particles within circular wedges that are lithographically shaped to apply a gravitational tilt to the particles toward the wedge's centre. However, this strategy does not allow to tune the strength of the confining trap as in our case, and it is difficult to apply to nanoscale particles where thermal fluctuations will easily facilitate particle escape from the lithographic traps. The third work does not use a circular confinement, but rather localizes the particles between coplanar thin electrodes using high frequency electric fields. In such case, while the confinement strength could be effectively tuned by the applied field, the final confinement is achieved only in one direction, leaving particles unrestricted (free) in the perpendicular direction. Furthermore, when considering several particles, the latter tend to form chains due to the presence of anisotropic, in-plane dipolar interactions. In our setup, interparticle interaction remains effectively isotropic. We comment on these works in the discussion part of the text by writing:

“We note also that trapping of microscale colloids has been achieved both by lithographic wedges [New Reference 1, New Reference 2] and high frequency electric fields [New Reference 3]. However, our work allows us to trap nanoscale magnetic particles via a magnetic landscape with

tunable stiffness, keeping them confined to a surface in stable two-dimensional clusters in absence of any topographic relief.”

[New Reference 1] P. Bahukudumbi and M. A. Bevan, J. Chem. Phys. 126, 244702 (2007).

[New Reference 2] G. E. Fernandes, D. J. Beltran-Villegas, and M. A. Bevan, J. Chem. Phys. 131, 134705 (2009).

[New Reference 3] J. J. Juarez and M. A. Bevan, J. Chem. Phys. 131, 134704 (2009).

The paper is well written and organized and the findings are generally supported by the results and their discussion.

Authors: We again thank the Referee for the positive feedback.

Response to Reviewers #3

Referee: *The authors present a nice study on magnetic trapping of nanoparticles in a magnetic field landscape. The study is clean, thorough and well executed. The experiments describe interesting regimes of tunable collective magnetic nanoparticle behaviour with the interparticle interaction strength tuned using the external magnetic field.*

Authors: We thank the Referee for all the positive comments on our manuscript.

Referee: *While the system has some degree of tunability in that the interaction strengths can be tuned using an external B field, the general properties of the landscape still seem hardwired into the system by assembly of the domain walls. Can the underlying domain wall structure be reconfigured at will - if so, could the authors comment on how and on what timescales? This would be an important feature to highlight that contrast the approach to nanofabrication-based trapping techniques where the landscape is largely hardwired by lithography as the authors mention.*

Authors: The magnetic bubble lattice can be easily controlled by an external field applied perpendicular to the film. Indeed, the zero-field equilibrium structure is a magnetic maze, which can be transformed into a bubble lattice by a high frequency oscillating field. As an example, we attach a real-time video for the Referee (Video_for_Referee.mpg) that shows such transformation. From the video it is clear that the bubbles can be moved and manipulated within the film, and they form a lattice only once the field is switched off. The lattice orientation and configuration can be further changed in time by the external field, or using an underlying permalloy pattern, as proposed in the past for the use of magnetic bubbles in data storage.

Regarding time scale by which a magnetic Bloch wall responds to the external field, usually ferrimagnetic iron garnet films allow exceptionally high domain wall velocities, which can reach speed of the order of ~km/s (see for example J. Phys.: Condens. Matter 33 075802 (2021) and reference therein). Thus, one can consider that their response to an external field is almost instantaneous if compared to the typical self-diffusion time of nanoparticles, which is of the order of milliseconds. We comment this point in the manuscript, by writing:

“The magnetic bubble lattice can be easily configured by an out of plane magnetic field and the corresponding bubble domains manipulated via small gradient when the field is still on. Further, given the fast response of the Bloch wall to magnetic field, with a propagation speed of the order

of ~ 1 km/s [New Reference], these features make this type of approach a promising candidate for magnetic based nanoscale trapping.”

[New Reference] K. H. Prabhakara et al. J. Phys.: Condens. Matter 33 075802 (2021)

Referee: *Also, what is the smallest effective domain size achievable for this type of system? This could be important to know as it will ultimately limit the maximum particle density in any possible high density array type of application.*

Authors: For this particular ferrite garnet film (FGF), which is characterized by ferromagnetic bubbles with a diameter of $D = 8.8$ micron in absence of applied field ($B_z = 0$ mT), we find experimentally that with a reverse field of $B_z = -10$ mT one can reach a diameter $D = 7$ micron. However, the film used is relatively thick (50 micron) and one can synthesize thinner FGF to reach much smaller size down to 500 nm as recently demonstrated [Büttner et al. Phys. Rev. Materials, 011401(R) (2020)]. The smaller magnetic bubbles could be easily manipulated by external field, the only disadvantage being the film thickness that requires special handling cares. We comment this point in the text by writing:

“Further, while our magnetic domains are relatively large, much smaller bubbles could be also synthesized in garnet film [New Reference] that could eventually lead to further system miniaturization.”

[New Reference] Büttner et al. Phys. Rev. Materials, 011401(R) (2020)

Referee: *A few minor points:*

It is not clear very clear what the term "enslaved" in the title really means. Do the authors possibly mean something along the lines of "confined"? The term is not mentioned or defined anywhere in the manuscript. If this is field specific jargon the suggestion is to use a different word/more common technical term or defining the term in the text for a broad interdisciplinary audience.

Authors: By the term “enslaved” we mean that the nanoparticles are confined within the magnetic domains and can only leave them under external command, i.e. when for example an external field is applied perpendicular to the FGF surface. This term does not belong to field specific jargon. However, to avoid any misunderstanding, we specify its meaning by writing in on page 7, first column:

“Thus, the trapped nanoparticles are enslaved within the magnetic bubble, and can leave them only by an external command, such as an applied field.”

Referee: *Towards the end of the manuscript the authors refer to the possibility of using their approach for storing and erasing information. Do the authors see any prospects for realising a magnetic colloid-based memory/data storage system along the lines of the electrostatic version described, e.g., in Myers et al., Nat. Nano. 10 (2015)? Some comments in this direction may help to flesh out possible technological applications of the technique.*

Authors: We thank the Referee for bringing to our attention also the work of Myers et al, where information storage and retrieval was obtained via combination of lithography, electrostatic

confinement, and an applied external field. Our idea for storing information within the magnetic bubbles is to confine the nanoparticle there by sedimentation, empty the bubbles with the perpendicular field that will send the nanoparticles toward the interstitial region, and later make some of them returning. One could in principle couple the FGF film with a permalloy pattern that would force only some nanoparticles to return to prescribed bubbles in the film. However, we do not exclude that one could couple the FGF with a lithographic structure and perform basic digital operations, similar to that demonstrated by Myers *et al.*, by using now magnetic fields instead of electric one. We thus follow the Referee suggestion, and include some comments along these lines on page 7, column 2 by writing:

“In particular, a recent work [New Reference] demonstrated the possibility of combining lithographic confinement, electrostatic levitation and external actuation to store and retrieve logic information from levitated nanoparticles. Since our FGF could be easily coupled with a lithographic structure, similar operations could be parallelized on the whole bubble lattice using an external magnetic field.”

[New Reference] C. J. Myers, M. Celebrano, and M. Krishnan, Nature Nanotech. 10, 886-891 (2015).

Reviewers' Comments:

Reviewer #1:

Remarks to the Author:

The authors have addressed and clarified all my concerns and the manuscript was revised accordingly. The differences of the present work with the previously published research are now clear and convincing. I congratulate the author for the excellent and exhaustive job, and I believe this already excellent work has gained further value from the revisions. I believe the authors have addressed the criticisms raised by the other reviewers, too. Thereby, in my opinion, the revised manuscript can be published as-is.

Reviewer #3:

None

Response to Reviewer comments

We thank the Referee for reading again our manuscript and supporting its publication in Nature Communications.

Referee: *The authors have addressed and clarified all my concerns and the manuscript was revised accordingly. The differences of the present work with the previously published research are now clear and convincing. I congratulate the author for the excellent and exhaustive job, and I believe this already excellent work has gained further value from the revisions. I believe the authors have addressed the criticisms raised by the other reviewers, too. Thereby, in my opinion, the revised manuscript can be published as-is.*

Authors: We are happy to have clarified all the issues. We thank the Referee for all the positive comments and for recommending the revised manuscript for publication.